# Understanding Breast Cancers through Spatial and High-Resolution Visualization Using Imaging Technologies

**DOI:** 10.3390/cancers14174080

**Published:** 2022-08-23

**Authors:** Haruko Takahashi, Daisuke Kawahara, Yutaka Kikuchi

**Affiliations:** 1Graduate School of Integrated Sciences for Life, Hiroshima University, 1-3-1 Kagamiyama, Higashi-Hiroshima 739-8526, Hiroshima, Japan; 2Department of Radiation Oncology, Graduate School of Biomedical Health Sciences, Hiroshima University, 1-2-3 Kasumi, Minami-ku, Hiroshima 734-8551, Hiroshima, Japan

**Keywords:** tissue clearing, optical three-dimensional imaging, spatial transcriptomics, photoacoustic imaging

## Abstract

**Simple Summary:**

Breast cancer tissue is not composed solely of cancer cells but exists in a complex microenvironment consisting of surrounding cells and proteins, including fibroblasts, immune cells, blood vessels, and extracellular matrix. The malignant transformation and metastasis of breast cancer occur due to the interaction between cancer cells and surrounding environmental cells and/or proteins. Therefore, it is extremely important to visualize the three-dimensional structure of breast cancer cells and their surrounding environment both invasively and noninvasively and to understand their relationship; this review aims to provide an overview of the tissue transparency techniques, optical observation methods, spatial transcriptomic analysis, and noninvasive medical imaging methods used for understanding malignant breast cancers.

**Abstract:**

Breast cancer is the most common cancer affecting women worldwide. Although many analyses and treatments have traditionally targeted the breast cancer cells themselves, recent studies have focused on investigating entire cancer tissues, including breast cancer cells. To understand the structure of breast cancer tissues, including breast cancer cells, it is necessary to investigate the three-dimensional location of the cells and/or proteins comprising the tissues and to clarify the relationship between the three-dimensional structure and malignant transformation or metastasis of breast cancers. In this review, we aim to summarize the methods for analyzing the three-dimensional structure of breast cancer tissue, paying particular attention to the recent technological advances in the combination of the tissue-clearing method and optical three-dimensional imaging. We also aimed to identify the latest methods for exploring the relationship between the three-dimensional cell arrangement in breast cancer tissues and the gene expression of each cell. Finally, we aimed to describe the three-dimensional imaging features of breast cancer tissues using noninvasive photoacoustic imaging methods.

## 1. Introduction

Breast cancer is the most common cancer affecting women [1,2]. Just as lung and colorectal cancers have been, breast cancer has also been the subject of numerous histological and molecular biological analyses [3,4]. Recent cancer research has focused on the analysis of the tumor microenvironment (TME), which includes not only the cancer cells themselves but also the surrounding cells, such as immune cells, fibroblasts, and mesenchymal cells [5,6]. The malignant transformation of cancer is regulated by the interaction between TME constituent cells and cancer cells [5,6,7]; however, the relationship between the location of cells within the cancer tissue and the malignancy of cancer is still poorly understood; moreover, one of the major reasons for the lack of understanding is the difficulty in identifying the three-dimensional (3D) location of each type of cell and tissue that constitutes cancer and the lack of progress in cellular-level analysis.

Imaging techniques in biology allow visualization from whole tissues, e.g., by immunohistochemistry (IHC), to the cellular and intracellular organelle level by direct labeling [8,9]. In particular, the recent advances in microscopic performance, staining methods, and cell/protein labeling techniques have made high-precision imaging possible [8,9]; however, as the imaging of deep tissues remains a challenge, it is still difficult to understand their 3D structure. Recent advances in tissue transparency and microscopy techniques have facilitated the imaging of the entire cancer structure, and analysis is underway to reveal the 3D structure and positional relationships of the cell types and tissues that contribute to the formation of cancer [10,11,12]. Recently, techniques have been developed to evaluate the gene expression of cells at specific locations in tissues, and data on the correlation between the location of cancer cells and their malignant transformation are accumulating [13,14,15].

This review aimed to provide an overview of the spatial and high-resolution visualization approaches using the latest imaging technologies such as tissue clearing methods, 3D imaging techniques, and integration of gene expression data in the cells to understand breast cancer (Figure 1). Medical imaging (computed tomography, magnetic resonance imaging [MRI], positron emission tomography [PET], and photoacoustic imaging) is a noninvasive method that allows 3D imaging of breast cancer tissues within the body (Figure 1). Although it is difficult to record images at the cellular level with these techniques owing to their limited resolution, they can collect information that cannot be obtained from biopsy or surgical specimens, and when combined with the information from the biological and molecular biological examination, they can provide a deeper understanding of the breast cancer tissue.

## 2. Tissue Clearing and Imaging

### 2.1. Principals and Methods of Tissue Clearing

Tissue transparency methods have been used for nearly 100 years; however, recent technological advances have increased the levels of transparency and, combined with IHC, have enabled observation at the cellular level [11,12,18]. In addition, recent improvements in microscopic technology have even made it possible to even visualize cells that are deep beneath the tissues. Such technological innovations allow the transparency analysis of not only normal tissues, such as the brain and internal organs but also of diseased tissues, such as cancers [11,12,18]. As a result, the 3D structure of the tumor tissue and the positional relationship of each tissue component (cancer cells, fibroblasts, immune cells, etc.) have been clarified.

Tissue transparency methods can be broadly classified into three categories: hydrophobic, hydrophilic, and hydrogel-embedding methods. More than 60 different clearing methods have already been reported [10,11,12,18]. Each transparency method uses different reagents and techniques, but the basic processes are the same: tissue fixation, permeabilization, decolorization, and refractive index (RI) matching. The principle of tissue transparency has long been studied; the RI of light is key to promoting tissue transparency [10,11]. Biological tissues are composed of lipids and proteins (RI: 1.5) that are immersed in a solvent called water (RI: 1.33). The difference in the RI between them causes the refraction, reflection, and scattering of light, which inhibits transparency [10,11]. Matching the RI to 1.33 or 1.5 using a transparency reagent, is the hydrophilic or hydrophobic clearing method, respectively. Although the clearing method using organic solvents is more efficient and requires less processing time than the other methods, few problems occur, such as sample shrinkage and fading of fluorescent proteins. By contrast, methods using water-soluble compounds are simpler and are characterized by less fading of fluorescent proteins. The third tissue transparency method, hydrogel embedding, is a rapid transparency method that involves the combination of hydrogel embedding and electrophoresis to remove large amounts of lipids and other proteins from the tissue. Owing to the complexity of the operation, a passive method that does not use electrophoresis but only involves immersion in a surfactant solution has been reported; however, it is time-consuming [10,11].

### 2.2. Application to Diseased Tissue Specimens

Large quantities of formalin-fixed paraffin-embedded (FFPE) samples are stored at medical institutions as patient-derived cancer tissues. The sections were prepared from these FFPE samples, and two-dimensional images were created using immuno-antibody staining and microscopic observation. Furthermore, a 3D image can be created from this two-dimensional image by volume rendering using a computer; however, as it is difficult to identify the structure of the entire cancer tissue in a two-dimensional image and conversion to a 3D image is labor intensive, a new method should be established to replace FFPE specimens.

The challenge associated with the application of tissue transparency in the clinical setting is its superiority in terms of tissue visualization against the existing hematoxylin and eosin (H&E) staining method. One of the most important questions that need to be addressed is whether this new method can be used in the medical field, as well as the H&E staining method, which has been used to determine various diseases. In order to test this issue, Nojima et al. performed a comparison between the clear, unobstructed brain/body imaging cocktails and computational analysis (CUBIC) staining method and the H&E staining method [19]. Tissue samples derived from patients with lymph node metastases were bisected and diagnosed using tissue transparency (CUBIC method), 3D imaging, and H&E staining. The results showed that the tissue transparency (CUBIC method)/3D imaging method was more sensitive for diagnosing of cancer metastasis [19].

### 2.3. Application to Breast Cancers

The 3D analysis of tissues by tissue clearing has been applied to a variety of diseased tissues; however, our review focuses on cancer tissues, especially breast cancer. Various tissue-clearing techniques have already been applied to several types of cancer tissues and have been used to analyze 3D tissue structures. In the case of breast cancer, the following mentioned studies have been published and have been applied to examine patients’ tissues, mouse xenograft tissues, and cancer organoids.

The female breast is composed of mammary glands and connective and fatty tissues, and the mammary gland consists of lobules that secrete milk and the branching ducts that are responsible for delivering the milk to the skin surface. Since breast cancer is known to begin primarily in the lobules and ducts [20,21], understanding the 3D structure of the mammary gland through biological imaging is important. Recent studies have shown that tissue clearing methods allow 3D imaging of the mammary glands: the virginal and lactating mouse mammary glands were visualized using the hydrophilic (see deep brain (seeDB) and CUBIC) and hydrophobic clearing methods (3D imaging of solvent-cleared organs (3DISCO)) [22]; moreover, the single-cell resolution 3D imaging of mouse and human mammary glands was achieved using the fructose, urea, and glycerol for imaging (FUnGI) method [23]. These tissue-clearing methods have been applied to acquire 3D images of the normal mammary glands and breast tumors [24], allowing analysis at the single-cell level [22,23]. The combination of labeling technology at the single-cell level, tissue clearing-3D imaging, and RNA-seq using transgenic mice has established a pipeline for resolving the heterogeneity of whole breast tumors [23].

One of the hydrogel-embedded, clear lipid-exchanged acrylamide-hybridized rigid imaging/immunostaining/in situ-hybridization-compatible tissue hydrogel (CLARITY) methods were developed and applied to the 3D imaging of intact adult mouse brains in order to examine the neural circuit wiring, cellular and subcellular structures, and localization of neurotransmitters [25]. Furthermore, biopsy specimens from breast cancer patients were analyzed using CLARITY for 3D imaging, and the findings were compared with those of 2D imaging using FFPE. Analysis of breast cancer specimens with CLARITY and 3D imaging improves the accuracy and enables the acquisition of unbiased analysis results compared with 2D imaging with conventional FFPE [16].

In addition to breast cancer tissues and biopsy specimens from patients, tissue-clearing technologies have also been applied to organoids and spheroids. Various human cancer organoids (airway, colon, kidney, liver, and breasts) and mouse mammary gland organoids have been examined using commercially available FocusClear™ (CelExplorer, Taiwan) or homemade fructose–glycerol clearing reagents, and 3D imaging of these organoids with a single-cell resolution has been achieved [26]. Another study showed that on-chip clearing methods for breast cancer spheroids on a microfluidics platform using the hydrophilic reagents (SeeDB, Clear^T2^, and ScaleSQ) were developed and will be used for drug screening and other applications [27].

### 2.4. Optical Imaging

Tissue clearing facilitates deep 3D imaging of thick tissues. The following three optical methods have been mainly used for 3D imaging of the interior of transparency-enhanced tissues: confocal laser microscopy, multi-photon microscopy, and light-sheet microscopy.

In conventional widefield fluorescence microscopy, fluorescence signals from the focal plane and outside the focal plane are detected, resulting in the acquisition of low-resolution images. On the contrary, confocal fluorescence microscopy improves the image resolution because the pinhole eliminates the signals outside the focal plane [28]. In addition, the laser was scanned and images of the focal plane were acquired along the z-axis, thus allowing imaging of the entire tissue [29]. Multiphoton fluorescence microscopy is based on the basic principle of two-photon excitation by ultrashort laser pulses and overcomes the disadvantages of confocal fluorescence microscopy (low tissue penetration and high invasiveness) [30,31]; this multiphoton fluorescence microscope uses long-wavelength excitation light, which has high tissue penetration and is suitable for live imaging [30,31].

The combination of multiphoton fluorescence microscopy and tissue clearing method further improves tissue penetration along the z-axis, but imaging of the whole tissue by 3D scanning requires time-consuming measurements, as well as volume rendering for 3D imaging [29]. To overcome these limitations, a light-sheet microscope was developed to enable large-scale, high-speed imaging [32,33]. Light-sheet microscopy achieves high-speed 3D imaging by creating optical sections in a sample with excitation light spread in the form of a sheet and collecting these 2D images as a continuous tomographic image [32,33]. Since light-sheet microscopes collect images from a 90-degree angle to the irradiation of the excitation light, the object to be observed must be transparent. Therefore, tissue transparency technology and light-sheet microscopy are used as a single set [32,33]. Light-sheet microscopy, when combined with tissue transparency techniques, provides a much deeper field of view compared with confocal microscopy or two-photon fluorescence microscopy; however, its spatial resolution is greatly compromised [29].

## 3. Spatial Transcriptomics

### 3.1. Transcriptomics Analysis in Single-Cell Resolution

Recent advances in next-generation sequencing technology, bioinformatics, and computational science have led to dramatic advances in omics analysis technology, which handles whole biological information at hierarchical levels such as the genome, epigenome, transcriptome, and proteome levels [34,35,36,37,38,39]. Furthermore, numerous algorithms and tools have been developed to analyze huge amounts of multi-omics data in an integrated manner; thus, the bioinformatics analysis techniques are more commonly used by cancer biologists [34,37,40,41,42]. In particular, transcriptomics obtained by RNA sequencing (RNA-seq), which quantitatively analyzes the RNA (mainly mRNA) expression in the whole genome, is useful as a basic source of cellular biological information that allows the detailed examination of cellular conditions. In breast cancer, transcriptome data have been collected from tumor and normal tissues and have been applied to analyze the characteristic gene expression profiles, called signatures or modules, and their relationship with the patient’s prognosis [43,44,45,46,47,48], as well as to identify the candidate biomarker genes and signaling pathways that are potential therapeutic targets [49,50,51,52]. To date, a worldwide effort is underway to collect transcriptomes of tissue samples from patients with breast cancer and other types of cancers; a vast amount of information has been accumulated in large databases such as The Cancer Genome Atlas (TCGA) (https://www.cancer.gov/about-nci/organization/ccg/research/structural-genomics/tcga, accessed date: 1 July 2022) led by the National Institute of Health (NIH) of the United States. As of 2022, the TCGA has access to 9116 breast cancer cases, both primary and metastatic, and 4737 RNA-seq BAM files (binary alignment map, a format for sequencing data).

In addition, single-cell RNA-seq (scRNA-seq), which uses microfluidics technology to obtain transcriptomics information of each single cell, has recently become available for general-purpose analysis, making it possible to deeply penetrate the heterogeneity of cell populations existing within a tissue [53,54,55]. Cancer tissue is composed of highly heterogeneous cells, not only including cancer cells but also the surrounding TME cells, which are closely related to the process of tumorigenesis and malignancy [56,57,58,59]. Breast cancer is classified into four classes (luminal A, luminal B, human epidermal growth factor receptor-2 (HER2), and triple-negative (TN)) according to the expression status of estrogen receptor (ER), progesterone receptor (PR), and HER-2, each with a different general prognosis and appropriate treatment strategy [60,61,62]. Of these subtypes, the TN subtype accounts for approximately 15% of all cases and has a higher probability of metastasis or recurrence and lower survival than the other three hormone- and HER2-positive subtypes, thus increasing its malignancy potential [63]. For the TN group, molecularly targeted drugs against receptors and related signaling pathways, as well as immune-targeted drugs, are less effective, and only chemotherapy with anticancer drugs has been established as an effective treatment, making it the subtype with the poorest prognosis [64]; moreover, even within the same class of patients, the degree of sensitivity or resistance to therapeutic anticancer drugs and the incidence of metastasis differ; hence, the heterogeneity of each patient cannot be ignored. The factors that can help determine this heterogeneity include the different characteristics and proportions of the cancer cell population, as well as their interaction with the surrounding TME cells [65]. As a recent example, when pairs of primary and metastatic tumor tissues were collected from TN patients and evaluated to determine their gene expression, gene mutations were rarely observed, but the gene expression signatures of immunomodulatory and other genes changed; this shift is thought to be one of the causes of the reduced or lack of efficacy of immunotherapeutic agents [66]. High-resolution profiling of cancer tissue at a single-cell level using scRNA-seq is a powerful tool to elucidate the nature and characteristics of the cell populations in complex types of cancer [67,68]; however, this method does not provide positional information about which cells a cancer cell is interacting with when the tissue is broken down into individual cells for sequencing.

### 3.2. Spatial Gene Expression Analysis of Cancer Tissue

In general, the clinical examination of breast cancer involves the use of staining methods such as IHC and fluorescence in situ hybridization (FISH) to visualize and evaluate the expression of each receptor (ER, PR, and HER2) in tissue sections, which are then classified into four classes. Additionally, several gene tests, such as Prosigna (PAM50), the Breast Cancer Index (BCI), and Oncotype DX, are used to support the outcome prediction and therapy strategy decisions [60,69,70,71,72,73,74]. PAM50 is a generic oncogene panel for breast cancer that measures the expression level of 50 cancer-related genes as a package using microarrays, etc., and classifies them according to their gene expression characteristics (gene signature) into five subtypes (luminal A, luminal B, HER2 enriched, basal-like, and normal-like). PAM50 is used for determining the patient’s prognosis. Interestingly, the positive/negative determination of receptor expression may differ between the conventional classification based on IHC or FISH staining of patient sections and the PAM50 classification based on the gene expression, although the classification does not always agree with the receptor expression status [75]. The staining intensity and mRNA expression levels differ; this might suggest that heterogeneity exists even within single cancer tissue. Oncogene tests, including PAM50, usually measure the tumor bulk; therefore, only averaged information from multiple cell populations is available.

Both gene expression information at the single-cell level and spatial information on the location of each cell could provide an essential profile of the cell population that makes up cancer tissue. Recently, several methods have been developed and proposed to obtain comprehensive biomolecular information (RNA, protein, etc.) using imaging-based cell-location tagging. For example, IHC and FISH can only stain a few target genes; however, imaging mass cytometry, which uses the time-of-flight (TOF) mass to read the information from metal isotope-labeled antibodies, has enabled the simultaneous staining of multiple samples [76,77,78]; this approach enabled the visualization of breast cancer and TME cells by staining 35 biomarkers simultaneously and pathologically classifying them into 18 subgroups by analyzing their architecture [79]. The spatial transcriptome has also been developed to obtain information on both spatial location and gene expression by reacting tissues with spatially barcoded mRNA-binding oligonucleotides as probes and detecting the probes bound to the mRNA [80]. Combined analysis of the spatial transcriptome with scRNA-seq data revealed the heterogeneity within tissues and interactions with the surrounding tissues. Based on the characteristics of the gene expression in breast cancer cells, it was possible to find more detailed gene signatures and gene modules compared with the previous four classifications as well as to extract the structural characteristics, including those of TME cells (see “Spacial Transcriptomics” in Figure 1) [17]. Based on the overall picture of the complexity of cancer tissues we discussed above, gene signatures to distinguish between cases with good and poor prognosis within four (or five) classifications have been discovered, and this could help explain the differences in prognosis among patients. The correlation between staining images obtained by IHC or FISH and PAM50 signatures obtained by bulk RNA-seq or microarrays was also discussed; more data will be collected in the future to enable more detailed subgroup identification and prognosis prediction based on the patient’s test results.

These spatial-gene expression analyses have captured the comprehensive picture and complexity of cancer at a high resolution, which were not found using previous methods, and have led to altered classifications; however, to date, these analyses have mainly focused on providing detailed descriptions and are yet to discover previously unknown relationships between cancer and cancer cells or between cancer and TME cells. In the future, results of integrated omics analysis at the single-cell level targeting the arbitrary spatial locations in cancer tissues will serve as a basis for conducting studies that will reveal new tumorigenesis and regulation mechanisms that have not been envisioned in the past.

## 4. Medical Imaging

### 4.1. Recent Advances of Medical Imaging for Breast Cancer

In clinical practice, X-ray mammogram, ^18^F-fluorodeoxyglocuse (FDG)-Positron Emission Tomography (PET), ultrasound (US) imaging, and Magnetic resonance imaging (MRI) are used for the screening and follow-up monitoring.

X-ray mammography can be used for mass screening in order to detect breast cancer. It is known as the gold standard technique for detecting breast cancer [81]. According to reports, breast cancer mortality reduced by 19% when using X-ray mammography for detection. However, the sensitivity of this technique decreases in dense breast tissue [82]. As an alternative screening method, MRI can be used for cancer screening in dense breasts [83]. MRI also provides detailed information such as staging, evaluation of microcalcifications and discharge, and premalignant lesions [84]. It has been clinically used to monitor the response to therapy and assessment of breast cancer recurrence and metastasis [85]. Although MRI usually requires a contrast agent, which can lead to allergic reactions in some patients. Additionally, MRI is more expensive than mammography: thus, it is not feasible as a mass screening procedure [86,87]. US also a major imaging techniques for monitoring the response to therapy and diagnosis of breast cancer [88]. Mammography and ultrasound are generally required for the diagnosis of breast pathologies [88]; it does not use radiation; thus, it is a powerful tool for use on pregnant and breastfeeding women to detect breast tumors; it has a high false-positive rate; therefore, it is often used in combination with mammography [89,90]. Computed tomography (CT) is potentially useful in patients with dense or intermediate-risk breast cancers. CT is less operator dependent than US imaging, and findings can be clearly localized in three-dimension. FDG-PET has been widely used in studies to determine the overall prognosis and aggressiveness of primary tumors [91,92]. FDG uptake correlates with the expression of hypoxia-inducible factor 1α [93], which is increased in the primary Ewing’s sarcoma family of tumors with distant metastasis [94]. However, it is difficult to differentiate malignant lesions from benign inflammatory processes or chronic hypoxia from normoxic tumors using only FDG-PET. Dynamic contrast-enhanced MRI (DCE) is a valuable method for assessing the microcirculatory environment in tumor tissues and providing information complementary to that of FDG-PET [95,96]. DCE-MRI is a valuable method for assessing the microcirculatory environment in tumor tissues [97,98] and providing information complementary to that of FDG-PET [99]. FDG-PET has high sensitivity (92–100%) for the detection of metastatic lymph nodes with a mixed specificity (77–93%) [100,101]. In contrast, a meta-analysis on the application of DCE-MRI reported a higher specificity than sensitivity for lymph node assessment [102]. The combination of MRI and PET is a promising approach to observe tumor biology in vivo [103] However, the feasibility of the combining PET and MRI for the assessment of tumor aggressiveness has not yet been investigated. Margolis et al. reported that FDG-PET and magnetic resonance pharmacokinetic parameters may aid in the assessment of metastatic potential and tumor aggressiveness [104] On the other hand, patients should fast for at least 4 to 6 h before the FDG-PET scan and patients take radiation exposure during the FDG-PET scan. US imaging is radiation-free and cost-effective but has a high false positive rate [89,105]. Therefore, a new imaging method which is not influenced by breast density, overcomes the above disadvantages is needed.

### 4.2. Photoacoustic Imaging

Recently, photoacoustic imaging combining optical excitation and ultrasonic detection has been proposed as a hybrid technique to address problems related to cost and breast density. The photoacoustic effect was caused by the thermoelastic expansion of biological tissues after irradiation with pulsed laser light, and generated pressure waves were detected using US transducers [106]. The generated pressure waves were detected using multiple US transducers. The acquired time of arrival and signal intensity we used to reconstruct the location and strength of the optical absorption in the tissue. Therefore, its resolution is much higher than that of the purely optical imaging modalities [107]. In addition, photoacoustic imaging is free of ionizing radiation because it uses near-infrared light. Photoacoustic imaging has different absorption peaks, which can distinguish different biological chromophores, such as water, hemoglobin, melanin, and lipids. The hemoglobin level in malignant tumors is higher than that in normal breast tissues [108]. Photoacoustic imaging focuses on the hemoglobin distribution to detect malignant tumors. Tumor angiogenesis requires nourishment to allow tumor growth [109]; it is a known biomarker for malignant tumors [110]. Breast cancers that grow over 2 mm in diameter can cause hypoxia due to capillary leaks and the formation of disorganized vascular structures [111,112]. Among the subtypes of breast cancers, triple-negative breast cancer (TNBC) shows aggressive characteristics with rapid growth and a higher recurrence rate [113]. Menezes et al. reported that TNBC tumors do not show external peripheral zone findings and rich internal findings on photoacoustic and US imaging, thus suggesting the possibility of differentiating TNBC from HER2-enriched breast cancer subtypes [114].

The PIONEER trial concluded that the combination of photoacoustic imaging and US imaging exceeded the specificity by 14.9% compared with that of internal US imaging [115]; moreover, the results could be made more robust by integrating elastography or US tomography [116,117]. Nyayapathi summarized the photoacoustic imaging system and the method of using this tool in breast cancer detection [118].

Photoacoustic imaging can be used for computer-aided detection and has been widely adopted in the field of radiology. Photoacoustic breast imaging has been rapidly advanced, and several in vivo studies have been performed [119]. These studies show the promise and potential of the photoacoustic method in the diagnosis and detection of breast cancer.

## 5. AI-Based Analysis of Spatial Transcriptomics and Medical Images

### 5.1. Spatial Transcriptomics and Artificial Intelligence (AI)

Nowadays, a number of types of “image” information, including images of sectioned tissues on slides, 3D images of tissue blocks, and medical images, have gone beyond the stage of being dealt as mere pictures and are now being digitized and processed by computers. These digitized images are big data with a huge breadth of information on both spatial and morphological characteristics, and it is beyond our human abilities to handle this enormous amount of information effectively and to extract and interpret meaningful information from it. Therefore, many attempts are currently reported to utilize artificial intelligence (AI) to solve these challenges (Figure 2).

Gene expression patterns acquired at the single-cell level and spatial transcriptomes have revealed heterogeneity that exists within breast cancer tissues of patients [65,67,68,77,79,80]. In order to utilize and further expand these findings, it is important to improve the accuracy by combining them with other modalities. Especially in the field of diseases such as cancers, pathological images (IHC, FISH, H&E staining, etc.) are very valuable because the vast accumulation of past data can be utilized and possibly applicable to clinical practical use [14,120]. Indeed, a prediction model of the spatial gene expression status from HE-stained images was developed by combining results of high-resolution gene expression acquisition using the spatial transcriptome with matched HE-stained histopathology images and learning it through deep learning techniques in breast cancer [121]. Additionally, a method for estimating spatial gene expression in tissues by deep learning bulk RNA-seq data and H&E-stained images was also reported [120]. Such deep learning results can be adapted to already publicly available data; thus, the value of using data collected in the past (before the generalization of the spatial transcriptome) has increased. In addition, although many studies have been reported, the regional size wherein the spatial transcriptome can be obtained in one cancer tissue is limited, and it is difficult to cover small invasive spots, for example. Likewise, it is difficult to detect and analyze low-expressing genes such as transcription factors. To approach information that is out of reach by only acquiring spatial transcriptomics, collecting the information together with tissue images by immunostaining or H&E and learning and integrating through deep learning allowed us to identify unknown regions and/or phenomena occurring in small areas of the tissue, which can further aid histological interpretation [122,123]. As another example, it may also be applicable to diagnostic support tools for cancers that are difficult to distinguish in clinical biopsies. Ductal carcinoma in situ (DCIS) and its precursor, invasive ductal carcinoma (IDC) are often difficult to distinguish from each other, but machine learning of the spatial transcriptome has revealed their respective gene expression signatures; hence, diagnosis for unlearned tissues that is consistent with the pathologist’s annotation with high accuracy was demonstrated [124].

As summarized above, by successfully integrating the different modalities of spatial transcriptomics and tissue staining images via AI-based learning, it is possible to expand the basic research area by discovering unknown biological features, as well as to implement this technology to practical use in hospitals, where clinical test specimens submitted daily will be diagnosed with high accuracy.

### 5.2. Medical Imaging and Artificial Intelligence (AI)

Artificial intelligence (AI) has been used to either complement the work of humans or replace them; it has an important role in image-recognition tasks and can be applied to clinical decision support and disease screening. AI methods mainly use machine learning and deep learning approaches. In the machine learning approach, the associated parameter and features are trained [125]. In the deep learning approach, a type of machine learning approach that constructs the layered architectures and extracts the features from simple to complex from the data [126] or the breast imaging, AI has been mainly used for the classification and detection of microcalcifications and tumors, density assessment, cancer risk assessment, and segmentation in mammography. The segmentation task generally used the deep learning approach. Wessam et al. compared various deep learning models (DenseNet121, InceptionV3, ResNet50, MobileNetV2, VGG16, and modified U-net models) for the breast cancer segmentation in mammography [127]. The accuracy reached 98.87% for the modified U-net model and 98.88% for the Inception V3 model. Some studies used a deep learning approach convolutional neural networks and You-Only-Look-Once. The quality detection accuracy was 98.96% [128,129]. For the assessment of breast density, the Breast Imaging-Reporting and Data System (BI-RADS), was machine-learning-based automated breast density software, that calculates the breast density and automatically outputs a report with the breast-density grade [130]. In the tumor detection and decision support tasks, a machine-learning-based cloud system for US imaging was developed [131]; it uses a supervised ML approach and supports physicians in the diagnosis of the region of interest [131]. The collaborating research group developed software that provides a decision-support tool with automated tumor detection of breast MRI images (http://gtr.ukri.org/projects?ref=104192, accessed date: 1 July 2022). Computer-aided detection (CAD) systems were introduced in 2011; however, it has been reported that the recall of CAD was increased due to no improvement in tumor detection rates [132]. Recently, the AI approach has been used for CAD. AI-Antari MA, et al. proposed a complete integrated CAD system for tumor detection, classification, and segmentation in X-ray mammography imaging. The accuracy was more than 92% for all tasks [133].

These applications and software need to be tested in the clinical environment as a prospective study. Therefore, controlled trials and cohort studies using large screening populations are warranted, which can contribute to understanding the potential changes in the performance of breast-screening methods using an integrated AI system.

## 6. Conclusions

In many types of cancer, including breast cancer, the cancer microenvironment is thought to be one of the factors contributing to resistance to cancer treatment. Due to the complex three-dimensional structure of breast cancer tissue, the effectiveness of treatments and drugs can vary even among breast cancer patients with the same genetic mutation. Therefore, understanding the three-dimensional structure of breast cancer tissue is important for future personalized medicine. In this review, we have outlined three topics for understanding the three-dimensional structure of breast cancer tissue: tissue clearing and imaging, spatial transcriptomics, and noninvasive medical imaging.

Currently, histochemical methods using FFPE samples are the mainstay for diagnosis in the medical field; this is because a system of analysis has already been established by which the status of breast cancer can be immediately ascertained by analyzing simple tissue sections. On the contrary, because tissue clearing and 3D optical imaging are time consuming and require skilled techniques, their application in the medical field is limited; however, with the advancement of medical sophistication and personalized medicine, it is expected that tissue transparency and three-dimensional imaging will be adapted to medical practice and that examining the three-dimensional structure of breast cancer tissue will become a routine examination. In addition to these 3D structures, analysis of the spatial transcriptomics and data on the efficacy and prognosis of therapeutic agents are expected to be accumulated. Once an AI system integrating these data is completed, it should be possible to select an appropriate treatment or therapeutic agent simply by inputting the spatial location of the cell types that make up breast cancer. Furthermore, if the correlation between medical images and 3D information of breast cancer tissue can be clarified by AI, it may be possible to infer breast cancer status from noninvasive medical images and assist in the selection of treatment methods. Collecting 3D information (images, genes, tissue information) and performing AI analysis should be the first step toward realizing personalized medicine.

## Figures and Tables

**Figure 1 cancers-14-04080-f001:**
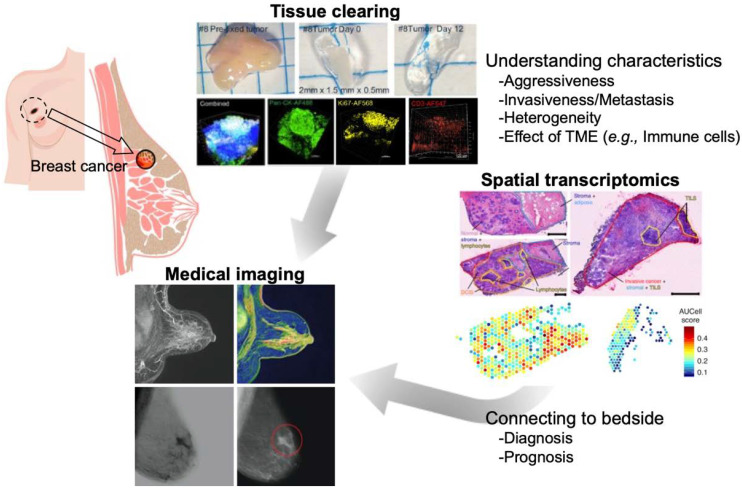
Spatial and high-resolution visualization using imaging techniques to understand breast cancer. Reprinted/adapted with permission from Ref. [16] (This article is an open access article distributed under the terms and conditions of the Creative Commons Attribution (CC BY) license (https://creativecommons.org/licenses/by/4.0/)) and [17], Copyright© 2022, Springer Nature America, Inc.

**Figure 2 cancers-14-04080-f002:**
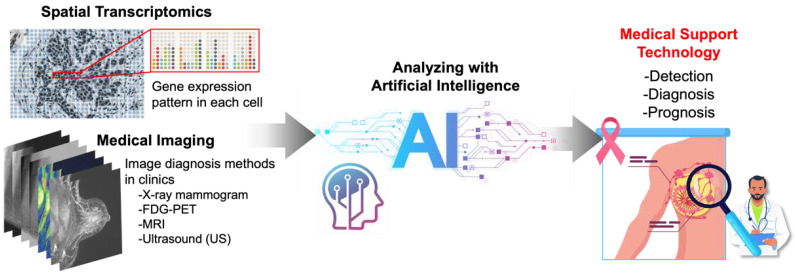
Artificial intelligence (AI) technology for spatial transcriptome data and medical image analysis, contributes to the development of medical support systems for practical applications.

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
