# Peer review of "Understanding Breast Cancers through Spatial and High-Resolution Visualization Using Imaging Technologies"

_cancers, 2022, doi:10.3390/cancers14174080_

Round 1

Reviewer 1 Report

Understanding Breast Cancers Through Spatial and High-Resolution Visualization Using Imaging Technologies

The paper presents an overview of the tissue clearing techniques, optical imaging methods, spatial transcriptomic analysis, and medical imaging methods used for understanding breast cancer malignancy. Overall, the problem is interesting and the paper is well written. However, some issues need to be fixed. To summarize:

Major strengths:
- Problem is interesting

- Paper is well-written

- References are adequate

Major weakness:

* The role of AI in understanding breast cancer malignancy is not clear

* Discussion is limited

Details comments:
1- Nowadays, AI plays a great role in interpreting the findings of medical imaging methods
 and spatial transcriptomic analysis to better understand breast cancer malignancy. The review needs to add a section to highlight its role. e.g., https://doi.org/10.1038/s41551-020-0578-x, https://doi.org/10.1038/s41598-022-07685-4,
https://doi.org/10.1016/j.ejca.2021.10.007, https://doi.org/10.1007/s12530-019-09297-2

2-Discussion is limited and can be improved by adding a section to present the findings of the paper and include the directions and trends for future work in the field

Author Response

Response to Reviewer 1 Comments

(In our revised manuscript, all revision points are highlighted in red.)

The paper presents an overview of the tissue clearing techniques, optical imaging methods, spatial transcriptomic analysis, and medical imaging methods used for understanding breast cancer malignancy. Overall, the problem is interesting and the paper is well written. However, some issues need to be fixed. To summarize:

Major strengths:

- Problem is interesting

- Paper is well-written

- References are adequate

Major weakness:

* The role of AI in understanding breast cancer malignancy is not clear

* Discussion is limited

Details comments:

1- Nowadays, AI plays a great role in interpreting the findings of medical imaging methods

 and spatial transcriptomic analysis to better understand breast cancer malignancy. The review needs to add a section to highlight its role. e.g., https://doi.org/10.1038/s41551-020-0578-x, https://doi.org/10.1038/s41598-022-07685-4, https://doi.org/10.1016/j.ejca.2021.10.007, https://doi.org/10.1007/s12530-019-09297-2

Answer: Based on the reviewer's advice, a new section “5. AI-based analysis of spatial transcriptomics and medical images” and Figure 2 were added to lines 387-465 to highlight the role of AI.

2-Discussion is limited and can be improved by adding a section to present the findings of the paper and include the directions and trends for future work in the field

Answer: We have revised the “6. Conclusions” in our manuscript to present the findings of the paper and include the directions and trends for future work in the field.

Reviewer 2 Report

The authors tried to present an overview of the imaging technique to assess biological information. Several references are missing especially on the medical imaging part. The review is too succinct to be representative to the whole imaging techniques. The authors should work on it. 

Major comments:

-          The English language should be strongly improved.

-          The medical imaging techniques should be more detailed. The review is too weak on this part. Moreover, there is several very important literature and advanced facts on imaging techniques missing, particularly on the ultrasound imaging technique and the CT (mammography). I don’t understand why the authors particularly focused on photoacoustic imaging, that is only one part of the recent advances on breast medical imaging techniques. If the authors want to be so succinct on the medical imaging techniques, they should change the title by ex vivo imaging techniques…

-          The review should include figures and results to illustrate the different part of the manuscript.

Minor comments:

-          Ln10: “A breast cancer tissue is composed of cancer cells”. It is obvious that cancer is composed of cancer cells. Please reformulate.

-          Ln 47, “Visualization techniques in biology not only utilize immunohistochemistry (IHC), hich allows visualization of the entire tissues, but also imaging techniques that directly label cells and intracellular organelles” is not totally true since some imaging techniques can also assess the perfusion of the tumor, the vascularization of the tumor,… and not only at cell level. The authors should reformulate this sentence.

Author Response

Response to Reviewer 2 Comments

(In our revised manuscript, all revision points are highlighted in red.)

The authors tried to present an overview of the imaging technique to assess biological information. Several references are missing especially on the medical imaging part. The review is too succinct to be representative to the whole imaging techniques. The authors should work on it. 

Major comments:

-   The English language should be strongly improved.

Answer: We revised the English language of our manuscript by using the expert English editing services recommended by Cancers. All English editing points are highlighted in red.

-    The medical imaging techniques should be more detailed. The review is too weak on this part. Moreover, there is several very important literature and advanced facts on imaging techniques missing, particularly on the ultrasound imaging technique and the CT (mammography). I don’t understand why the authors particularly focused on photoacoustic imaging, that is only one part of the recent advances on breast medical imaging techniques. If the authors want to be so succinct on the medical imaging techniques, they should change the title by ex vivo imaging techniques…

Answer: Based on the reviewer's advice, we have added the detailed medical imaging techniques (lines 310-313; 315-318; 320-327; 345-352; 381-385).

-          The review should include figures and results to illustrate the different part of the manuscript.

Answer: Based on the reviewer's advice, we have added Figure 2 to illustrate the different part of the manuscript.

Minor comments:

-          Ln10: “A breast cancer tissue is composed of cancer cells”. It is obvious that cancer is composed of cancer cells. Please reformulate.

Answer: Based on the reviewer's advice, we have revised a line 10 sentence.

-          Ln 47, “Visualization techniques in biology not only utilize immunohistochemistry (IHC), hich allows visualization of the entire tissues, but also imaging techniques that directly label cells and intracellular organelles” is not totally true since some imaging techniques can also assess the perfusion of the tumor, the vascularization of the tumor,… and not only at cell level. The authors should reformulate this sentence.

Answer: Based on the reviewer's advice, we have revised a line 47 sentence.

Round 2

Reviewer 1 Report

The authors have addressed the comments

Reviewer 2 Report

The authors took seriously the different comments. The paper can be accepted.